# Modeling the joint distribution of firm size and firm age based on grouped data

Chen Ge[1], Shu-Guang Zhang[1], Bin Wang[2]*

1 Department of Finance and Statistics, University of Science and Technology of China, Hefei, Anhui, China,
2 Department of Mathematics and Statistics, University of South Alabama, Mobile, AL, United States of America

* bwang831@gmail.com

## Abstract

The firm size distribution is highly skewed to the right and often follows a power law. In practice, it is common that firm size and firm age data are aggregated and released as grouped data to avoid disclosure of confidential information. We investigate multiple parametric methods for firm size and firm age modeling based on grouped data, and propose to estimate the joint distribution of firm size and firm age using the Plackett copula. The goodness-of-fit of the estimated marginal distributions are benchmarked with respect to the fit to the whole data and to the upper tails, respectively. The utilization of the proposed methods are demonstrated via an application to the 1977-2014 US firm data. Results show that the generalized lambda distribution has overall better performance in modeling both firm size and firm age data. The exponentiated Weibull distribution also works well in modeling the firm size data. As a by-product, the estimated parameter of the Plackett copula provides a measure of the association between firm size and firm age.

**Data Availability Statement:** The data can be downloaded from the webpage of Census's Bureau via https://www.census.gov/programs-surveys/susb.html.

**Funding:** The author(s) received no specific funding for this work.

## Introduction

Firms of different sizes and ages play different roles on employment, innovative activities, economic growth and other aspects of social and economic life. It was reported that firm size and age have a negative effect on firm growth for manufacturing firms [1–5], mining, wholesale and retail firms [6], and firms in the fields of construction, trade and service industries [7–10]. However, Audretsch et al. discovered different relationships in wholesale and hospitality industries in The Netherlands [11]. It was also believed that smaller and younger firms are fundamental to job creation and growth as they bring new products to market and promote growth through competition. Meanwhile Pagano and Shivardi revealed that large size is associated with faster productivity growth by investigating the relationship between size distribution of firms and sectoral productivity growth in the 1990s based on a Eurostat (1998) database [12, 13]. By examining the evolution of firm size and employment share distribution in Japanese and UK manufacturing between 1972 and 1991, Doi and Cowling found significant and important differences between the two countries. In the UK, the small firms provided increase share of both the total stock of firms and employment, while in Japan the small firms didn't

**Competing interests:** The authors have declared that no competing interests exist.

increase the employment share over the period [14]. Yasuda investigated the impacts of firm size, firm age and firm behavior, specifically R&D activities and subcontracting, on firm growth and survivability by fitting a multiple regression model based on the data of nearly 14,000 Japanese manufacturing firms extracted from the "MITI Basic Survey of Business Structure and Activities (SBSA)" survey [15]. He found that firm size and firm age have negative effects on firm growth, and they have positive effects on a firm's survivability.

The estimation of firm size distribution can be traced back to the late 1900's [16]. Literature shows that the size distribution of firms conforms fairly well to the log-normal distribution, with possibly some skewness to the right [17–19]. Though, Quandt tested for firm size distribution and rejected the log normality for the Fortune 500 in both 1955 and 1960. He found that the Pareto distribution fits better than the log-normal distribution [20]. Gao et al. investigated the temporal evolution of the size distribution of China's listed companies by modeling the upper tail behaviors using a Pareto distribution [21]. Cirillo and Hüsler analyzed the upper tail of the size distribution of Italian companies with limited liability belongings to the CEBI database and found that the largest firms follow a power law distribution. The power law hypothesis was also positively tested using graphical and analytical methods [22]. Segarra and Teruel analyzed the effect of sample size on the firm size distribution of Spanish manufacturing firms for the years 2001 and 2006 and showed the existence of a non-constant power-law distribution. Further they discovered that the firm size distribution of employment is more sensitive to firm age than that of sales [23]. Clementi and Gianmoena modeled the dependence structure between income and consumption parametrically using the "symmetrized Joe-Clayton" copula [24].

Different variables have been used to measure the firm size, including number of employees, revenue, net worth, sales, among many others [25–33]. In practice, published firm size (and age) data are often heavily aggregated, which poses big challenges to the density/distribution estimation of firm size or firm age, or their joint distributions, due to information loss. Furthermore, the grouped firm size (and age) data are often top-coded which makes non-parametric distribution estimation challenging and parametric approaches with moment matching method infeasible [34]. In this study, we use the number of employees to measure the firm size, and assume that both the firm size and firm age data have been aggregated into a small number of classes. The marginal distributions of firm size and firm age are estimated by fitting the grouped data to various well-known families of distributions. Specifically, four distributions including the log-normal distribution, the Pareto (type I) distribution, the generalized Pareto distribution, and the generalized lambda distribution are used to model the firm size data. The firm age data are modeled using the exponential distribution, the Weibull distribution, the exponentiated Weibull distribution, and the generalized lambda distribution, respectively. The Bayesian information criteria is used to choose the best fits of the marginal distributions, which will be further used to estimate the joint distribution of firm size and firm age using the Plackett copula [35].

## Materials and datasets

### Legacy BDS firm size and age data

In the US, data on employers are produced annually by the U.S. Census Bureau in the Statistics of U.S. Businesses (SUSB) program. The SUSB's employer data contain the number of firms, number of establishments, employment, and annual payroll for employment size of firm categories by location and industry. For illustration purposes, firm size and firm age datasets from 1977-2014 are selected from the annual measures of business dynamics for economy provided by the U.S. Census Bureau's Business Dynamics Statistics (BDS) program

**Table 1. Frequency distribution of 2014 US private-sector firms by size and age.**

|  | 1-4 | 5-9 | 10-19 | 20-49 | 50-99 | 100-249 | 250-499 | 500-999 | 1,000-2,499 | 2,500-4,999 | 5,000-9,999 | 10,000+ |
|---|---|---|---|---|---|---|---|---|---|---|---|---|
| 0 | 307192 | 51571 | 25777 | 13519 | 3715 | 1584 | 356 | 116 | 59 | 10 | 2 | 1 |
| 1 | 229474 | 43723 | 22922 | 12311 | 2844 | 980 | 185 | 69 | 46 | 10 | 6 | 23 |
| 2 | 184923 | 47678 | 24945 | 13551 | 3115 | 1056 | 220 | 75 | 44 | 15 | 8 | 27 |
| 3 | 153375 | 45156 | 24398 | 13507 | 3348 | 1299 | 268 | 123 | 59 | 10 | 7 | 20 |
| 4 | 126917 | 39706 | 22107 | 12588 | 3180 | 1413 | 328 | 129 | 59 | 22 | 6 | 19 |
| 5 | 119916 | 38401 | 22076 | 12634 | 3265 | 1206 | 273 | 109 | 67 | 21 | 6 | 19 |
| 6-10 | 558721 | 188947 | 108484 | 61942 | 16490 | 7112 | 1691 | 703 | 303 | 83 | 42 | 104 |
| 11-15 | 361374 | 136588 | 82244 | 49530 | 14376 | 6951 | 1712 | 762 | 406 | 140 | 73 | 71 |
| 16-20 | 260952 | 105546 | 64354 | 39669 | 11557 | 5507 | 1498 | 710 | 398 | 119 | 48 | 83 |
| 21-25 | 179719 | 79116 | 49055 | 31115 | 9124 | 4727 | 1320 | 588 | 343 | 125 | 63 | 82 |
| 26+ | 271235 | 135860 | 87894 | 62803 | 22264 | 13968 | 4733 | 2327 | 1426 | 555 | 291 | 276 |
| Pre-1977 | 113953 | 78743 | 63856 | 57303 | 25326 | 19587 | 7042 | 3896 | 2664 | 1081 | 654 | 821 |

[36]. The BDS data is a longitudinal database of business establishments and firms with coverage starting in 1976. Table 1 shows the firm size (in columns) by firm age data (in rows) for the 2014 US private-sector firms. Table 1 uses the actual firm size, which is defined as the average employment of the year of operation and the previous year. Firm age is computed for all firms in the Longitudinal Business Database (LBD) from the age of the establishments belonging to that particular firm, and the establishment age is computed by taking the difference between the current year of operation and the year the establishment first reports positive employment in the LBD. Both the firm size and firm age data have been aggregated into 12 classes and are top-coded.

The firm age data and firm size data from 1977 to 2014 were released on September 6, 2016 as a part of the "Legacy BDS Firm Characteristics Data Tables 1977-2014". More firm size and age data have been released for 2015 and 2016 in the BDS firm and establishment data tables.

## Methods

### Marginal distribution estimation by fitting parametric models to grouped data

Let $X$ be a random sample of size $n$ from a continuous distribution with density function $f$ and distribution function $F$, where the functional forms of $f$ and $F$ are assumed to be known with a vector of unknown parameters, $\theta$. Assuming that $X$ has been grouped into $k$ classes with class boundaries $0 \leq b_0 < b_1 < b_2 \ldots < b_k$. To fit $F$ to the grouped data, we compute the maximum likelihood estimates (MLEs) of the unknown parameters, $\hat{\theta}$, by maximizing the following log-likelihood function,

$$\ell(\theta) = \sum_{i=1}^{k} m_i \log[F(b_i) - F(b_{i-1})], \tag{1}$$

where $m_i \geq 0$ is the frequency of the $i$-th class satisfying $\sum m_i = n$. By adopting the maximum likelihood method, we can simply take $F(b_k) = 1$ when $b_k \to \infty$, and thus the difficulty caused by the infinite upper limit of the last class can be easily resolved. More details on the parameter estimation will be provided for the specific distributions later.

## Goodness-of-fit assessment

To assess the goodness-of-fit of a fitted distribution, $\hat{F} = F(x; \hat{\theta})$, to the grouped data, the following two methods are used:

1. $D_n$: the Kolmogrov-Smirnov (KS) statistic.

   Let $\hat{F}_n$ be the empirical distribution function (EDF), which approximates $F(x)$ using the proportion of observations in sample $X$ that are smaller than or equal to $x$. For grouped data, we approximate $F(x)$ at the boundary points as

   $$\hat{F}_n(b_i) = \frac{1}{n+1} \sum_{i=1}^{n} I_{X_i \leq b_i} = \frac{1}{n+1} \sum_{j=1}^{i} m_j, \tag{2}$$

   where $I_{X_i \leq b_i}$ is an indicator function assuming value 1 if $X_i \leq b_i$ and 0 otherwise. When $n$ is large, which is often the case for firm size and firm age data, $\hat{F}_n$ provides a very good approximation to $F$.

   To measure the goodness-of-fit of $\hat{F}$ to the underlying distribution function, we adopt the Kolmogrov-Smirnov statistic and compute the maximum distance between $\hat{F}_n$ and $\hat{F}$ over the class boundary points as

   $$\hat{D}_n = \max_i |\hat{F}_n(b_i) - F(b_i, \hat{\theta})|. \tag{3}$$

2. $\tilde{D}_n$: a measure of goodness-of-fit to the upper tail of $F$.

   The Zipf plot is a powerful graphical tool to visualize the upper tail behaviors of right-skewed distributions. It has been used to study the upper tail of the size distribution of firms [37, 38]. Let $X_{(i)}$ be the $i$-th order statistic. When the raw data are available, the rank in descending order of $X_{(i)}$ can be computed as

   $$R_i = n[1 - F(X_{(i)})] \approx n[1 - F_n(X_{(i)})]. \tag{4}$$

   The plot of $\log(R_i)$ against $\log(X_{(i)})$ is the so-called Zipf plot of $X$. For grouped data, we compute the rank of a boundary point $b_i$ as $r_i = n[1 - F_n(b_i)]$, and the Zipf plot of the grouped data can be obtained by plotting the $\log(r_i)$ against the $\log(b_i)$, for $i = 1, 2..., k$. Under $\hat{F}$, the rank of $b_i$ can be estimated as $\hat{r}_i = n[1 - F(b_i, \hat{\theta})]$ and we can obtain the Zipf plot of the fitted distribution by plotting $\log(\hat{r}_i)$ against $\log(b_i)$ [39]. To measure the goodness-of-fit of the fitted distribution $\hat{F}$ to the data, we compute the maximum distance in y-axis between the Zipf plots of the data and $\hat{F}$ as

   $$\tilde{D}_n = \max_i |\log(r_i) - \log(\hat{r}_i)|. \tag{5}$$

   $\tilde{D}_n$ is actually a KS statistic to measure the distance between two survival functions, $S_n(t) = 1 - F_n(t)$ and $\hat{S}(t) = 1 - \hat{F}(t, \hat{\theta})$ but at log-scales. $\hat{D}_n$ measures the goodness-of-fit of $\hat{F}$ to the whole data, while $\tilde{D}_n$ measures the goodness-of-fit of the fitted distribution to the upper tail of the data. If the data is very skewed to the right, $\hat{D}_n$ can be dominated by the first few classes where the data is very dense. A fitted distribution with small $\hat{D}_n$ does not necessary have small $\tilde{D}_n$.

## Fitting GLD to grouped data

The Tukey-Lambda distribution is a two-parameter distribution which specifies a distribution through a quantile function. It has various multi-parameter extensions [40–44]. The FMKL-GLD

(GLD hereafter) is one parameterization of the generalized lambda distribution family [45]. It specifies a distribution through the following function:

$$Q(u) = \lambda_1 + \frac{1}{\lambda_2}\left[\frac{u^{\lambda_3} - 1}{\lambda_3} - \frac{(1-u)^{\lambda_4} - 1}{\lambda_4}\right], \quad 0 \le u \le 1, \tag{6}$$

where $Q(.) = F^{-1}(x)$ is a quantile function. The GLD has four parameters: $\lambda_1$ is the location parameter, $\lambda_2$ is the scale parameters, $\lambda_3$ and $\lambda_4$ are two shape parameters. The GLD is very flexible and rich in shapes and tails. It can approximate many well-known distributions including the lognormal distribution, extreme value distributions, Pareto distribution among many others. The GLD can be used for statistical inferences of quantiles with very high or low levels [34, 46].

To fit GLD to grouped data, the moment matching and L-moment matching methods won't work well because of the missing of exact location information of the observations in the raw data due to binning. When the data are top-coded, it becomes challenging to well approximate the moments. Though, we can fit the GLD using the percentile matching method. When raw data are available, the following five percentiles, $p_u$ with $u = 10, 25, 50, 75, 90$, are used to estimate the unknown parameters by constructing a four-dimensional equation system. For grouped data, we can hardly obtain the five percentiles with the specific levels. For example, when the relative frequency of the first class is larger than 50%, we cannot precisely estimate $p_{10}, p_{25}$ (the lower quartile) and $p_{50}$ (the median) based on the grouped data. However, when we have six or more classes, we can use any five of the inner boundaries and their corresponding percentile ranks to build the equation system for parameter estimation.

The burden of computing the MLE of a GLD based on raw data is very heavy, as there is no closed-form solution for the distribution function $u = F(x)$ for given $x = Q(u)$ in (6). However, for grouped data, we have only a few boundary points where the values of the distribution function and the density function need to be evaluated, which greatly reduces the computational burden so numerically searching for the MLE becomes feasible. We use the percentile matching method to obtain initial estimates of the GLD parameters, then searching for the MLEs numerically by maximizing the log-likelihood function in (1). To avoid over-fitting and make the algorithm more efficient, we adopt a two-stage algorithm by first searching in the two-dimensional parameter space for $(\lambda_3, \lambda_4)$, then search for the best estimates for $(\lambda_1, \lambda_2)$ based on $(\hat{\lambda}_3, \hat{\lambda}_4)$. We also pose constraints $\lambda_3 > 0$ and $\lambda_1 \cdot \lambda_2 \cdot \lambda_3 > 1$ to ensure $f(x) = 0$ for $x \le 0$. More details can be found in [47, 48].

## Estimating the joint distribution of firm size and firm age

Let $X$ and $Y$ be two continuous random variables of the firm age and firm size, respectively. Denote the density and distribution functions of $X$ and $Y$ by $f_X$, $F_X$, $f_Y$ and $F_Y$, respectively. Also let $f_{XY}$ and $F_{XY}$ be the density and distribution functions of the bivariate distribution of $(X, Y)$. According to Sklar's theorem [49], $F_{XY}$ can be estimated as

$$F_{XY}(x, y) = C(F_X(x), F_Y(y)), \tag{7}$$

where $C$ is a copula. We adopt the Plackett copula and estimate the density and distribution functions of the bivariate distribution of $X$ and $Y$ as

$$f_{X,Y}(x, y) = \frac{\Psi f_X(x) f_Y(y)[1 + (\Psi - 1)(F_X(x) + F_Y(y) - 2F_X(x)F_Y(y))]}{(S^2(x, y) - 4\Psi(\Psi - 1)F_X(x)F_Y(y))^{3/2}}, \tag{8}$$

$$F_{X,Y}(x, y) = \begin{cases} \frac{S(x,y) - \sqrt{S^2(x,y) - 4\Psi(\Psi-1)F_X(x)F_Y(y)}}{2(\Psi-1)} & \Psi \neq 1 \\ F_X(x)F_Y(y) & \Psi = 1 \end{cases}, \qquad (9)$$

where $S(x, y) = 1 + (F_X(x) + F_Y(y))(\Psi - 1)$ and $\Psi > 0$ is an unknown parameter. If $X$ and $Y$ are independent, $\Psi = 1$. When $X$ and $Y$ are negatively correlated, $\Psi < 1$, and when $X$ and $Y$ are positively correlated, $\Psi > 1$. $\Psi$ can be estimated based on the joint frequency distribution table in Table 1 using the Plackett's method [35, 50]. The optimal estimate of $\Psi$ is found by maximizing the following likelihood function:

$$\tilde{\ell}(\Psi) = \sum_{i=1}^{K}\sum_{j=1}^{K'} m_{ij} \log P((X_i, Y_j) \in C_{ij}), \qquad (10)$$

where $C_{ij}$ denote the cell in the $i$-th row (age) and $j$-th column (size) in Table 1 and $m_{ij}$ is the corresponding frequency. $P_{ij} = P((X_i, Y_j) \in C_{ij})$ is the probability that a firm with size $X = x$ and age $Y = y$ falls within cell $C_{ij}$. It can be computed as follows:

$$\begin{aligned} P_{ij} &= P(b_{i-1} < X \leq b_i \text{ and } b'_{j-1} < Y \leq b'_j) \\ &= F_{X,Y}(b_i, b'_j) - F_{X,Y}(b_{i-1}, b'_j) - F_{X,Y}(b_i, b'_{j-1}) + F_{X,Y}(b_{i-1}, b'_{j-1}), \end{aligned}$$

where $b_{i-1}$ and $b_i$ are the lower and upper boundaries of the $i$-th class for firm age, and $b'_{j-1}$ and $b'_j$ are the lower and upper boundaries of the $j$-th class for firm size. We fix the parameters for $F_X$ and $F_Y$ and numerically search for $\Psi$ over a fine grid over $(a, b)$. To find the values of $a$ and $b$, we divide the joint distribution of $X$ and $Y$ into four quadrants using lines $x = b_i$ and $y = b'_j$, where $b_i$ and $b'_j$ are any combination of the inner boundary points of the classes for firm age and firm size. Let $N_k$, $k = 1, 2, 3, 4$, be the total number of firms in the $k$-th quadrant. The Plackett's estimator can be computed as

$$\hat{\Psi}_{ij} = (N_1 \cdot N_4)/(N_2 \cdot N_3), \quad \text{for } N_1, N_2, N_3, N_4 > 0.$$

Rough estimates of $a$ and $b$ can be found as $a = \min(\hat{\Psi})$ and $b = \max(\hat{\Psi})$.

## Results

### Fitting firm age distributions

For the firm age data in Table 1, $k = 12$ and we take $b_0 = 0$ and $b_k = \infty$. That says, a firm in class "0" is formed in the year of operation and can have an age greater than 0 but less than one year. Table 2 shows the class limits for firm age. For each class, the lower and upper boundaries are the same as the lower and upper limits, respectively. It is worth pointing out that the boundary between class "26+" and "Pre-1977" depends on the year of operation. For instance, there are only two classes with positive counts, "0" and "Pre-1977", for the year 1977 firm age data. For year 1978, three classes have positive counts: the firms established in 1978 with $0 < X < 1$, the firms established in 1977 with $1 \leq X < 2$, and those established before 1977 with $2 \leq X < \infty$, and so on.

**Table 2. Firm age class limits.**

| Class | 0 | 1 | 2 | 3 | 4 | 5 | 6-10 | 11-15 | 16-20 | 21-25 | 26+ | Pre-1977 |
|---|---|---|---|---|---|---|---|---|---|---|---|---|
| Limits | (0,1) | [1,2) | [2,3) | [3,4) | [4,5) | [5,6) | [6,11) | [11,16) | [16,21) | [21,26) | [26,38) | [38,∞) |

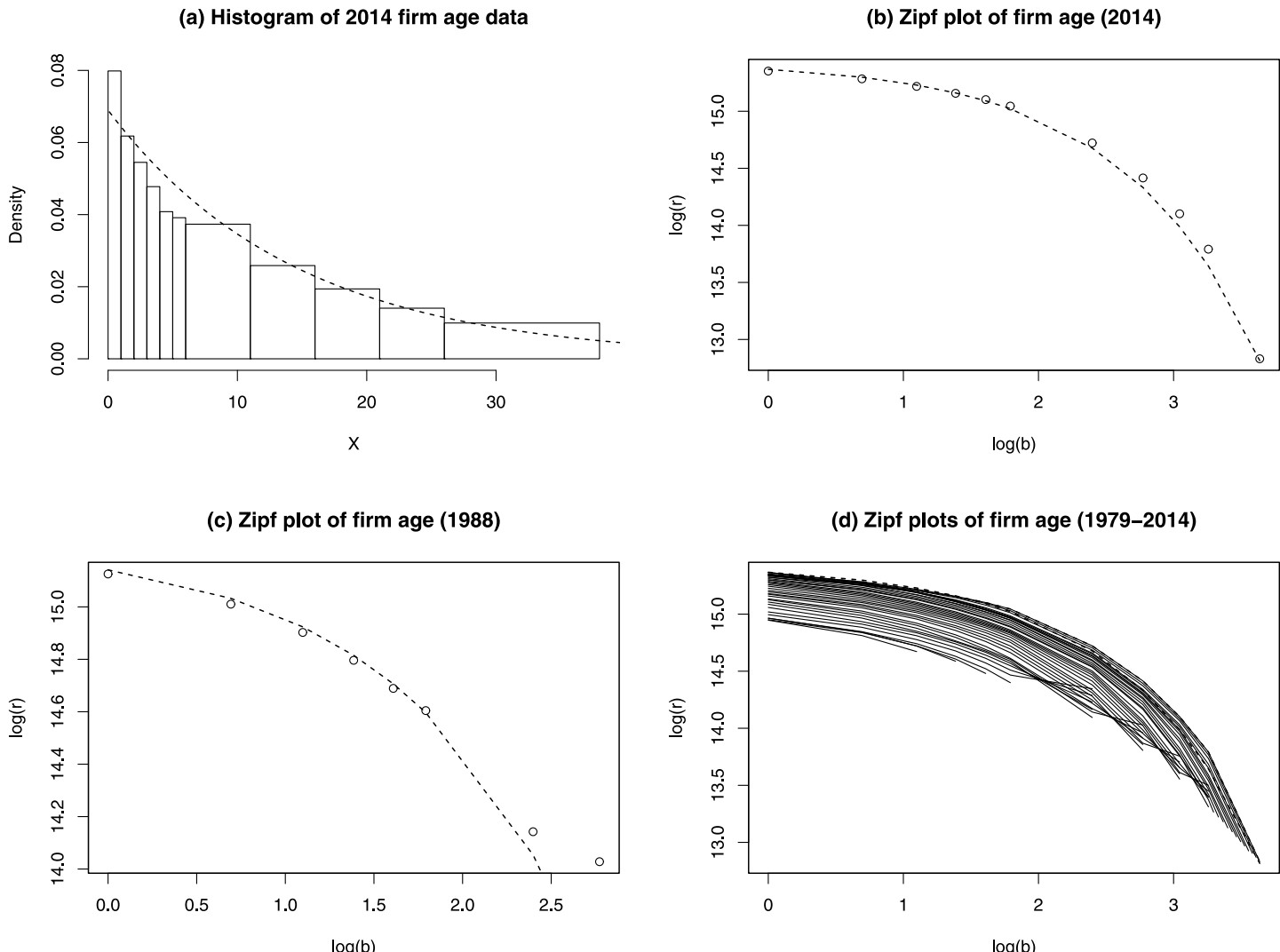

**Fig 1. US private-sector firm age data.** (a) Histogram of 2014 firm age data. The last class is top-coded and is not shown. The dashed curve shows the fitted exponential density function. (b) Zipf plot of 2014 firm age data (empty dots) and Zipf plot of the fitted exponential distribution (dashed curve). (c) Zipf plot of 1988 firm age data (empty dots) and Zipf plot of the fitted exponential distribution (dashed curve). (d) Zipf plots of firm age data for years from 1977 to 2014.

Plot (a) of Fig 1 shows the histogram of the 2014 US firm age data.

The firm age data is right-skewed. The dashed curve in plot (a) shows the density function of the fitted exponential distribution (EXP). Although the densities at the first class may be under-estimated, the EXP fits the overall firm age data pretty well. Plot (b) also demonstrates that the EXP fits the upper tail of the 2014 firm age data very well. Plot (c) shows the Zipf plots of the 1988 firm age data and the Zipf plot of the fitted EXP. We see that the EXP well fits the data for most of the classes but not the two or three classes at the upper tail. The Zipf plots of the firm age data for years from 1979 to 2014 are displayed in plot (d). It shows a clear pattern of parallel curves, which indicates that the firm age distributions for different years have similar upper tail behaviors and the EXP is a reasonably choice to be used to estimate the marginal distribution of firm age.

To accommodate more shapes and tail behaviors, we fit the firm age data to three other families of distributions that are more flexible than the EXP, namely, the two-parameter

Weibull distribution (WD), the three-parameter exponentiated Weibull distribution (EWD), and the four-parameter generalized lambda distribution (GLD). The distribution functions of the EXP, the WD and the EWD are given below,

$$
\begin{aligned}
F_{EXP}(x; \lambda) &= 1 - e^{-x/\lambda} \\
F_{WD}(x; \kappa, \lambda) &= 1 - e^{-(x/\lambda)^{\kappa}}, \\
F_{EWD}(x; \kappa, \lambda, \alpha) &= [1 - e^{-(x/\lambda)^{\kappa}}]^{\alpha},
\end{aligned}
$$

where $\lambda, \kappa, \alpha > 0$ are the unknown parameters to be estimated based on the grouped data. The EXP is a special case of WD when $\kappa = 1$. It has only one parameter and can fit data of limited shapes. WD is more general and can fit data of more shapes and tails. EWD is even more general than WD, which is equivalent to WD when $\alpha = 1$. When $\kappa = 1$, EWD is also the exponentiated exponential distribution, which will not be discussed in this study.

If the grouped data do come from an EXP, we have

$$
\lambda = -\frac{1}{b_i} \log \left[ 1 - \hat{F}_n(b_i) \right], \quad \text{for } i = 1, 2, \ldots, k-1, \tag{11}
$$

and therefore, we can find a rough estimate of $\lambda$ for EXP as

$$
\hat{\lambda}_{EXP} = -\frac{1}{k-1} \sum_{i=1}^{k-1} \frac{1}{b_i} \log \left[ 1 - \hat{F}_n(b_i) \right]. \tag{12}
$$

The MLE of $\theta = \lambda$ can be found numerically by maximizing the log-likelihood in (1), and use $\hat{\lambda}_{EXP}$ as the initial estimate.

For WD, we have

$$
Z_i = \log[-\log(1 - F(b_i))] = \kappa \log(b_i) - \kappa \log(\lambda), \quad \text{for } i = 1, 2, \ldots, k-1. \tag{13}
$$

We can fit a simple linear regression model of $Z_i$ on $\log(b_i)$ and obtain initial estimates of $\kappa$ and $\lambda$ using the slope and y-intercept of the fitted least squares line. These initial estimates are used to find the MLE by maximizing the log-likelihood function in (1).

To avoid search for the MLEs of EWD in a three dimensional space, we search for $\alpha$ over a fine grid around 1.0, say $(-5, 5)$, then fix the $\alpha$ value and do a power transformation as $\hat{F}_n(b_i)^{1/\alpha}$, for $i = 1, 2, \ldots, k-1$, then search for the MLEs of $\lambda$ and $\kappa$ using the same algorithm as for WD.

As an example, Fig 2 shows the results of fitting the EXP, WD, EWD and GLD to the 2014 US private-sector firm age data. The left panel shows that all four estimates provide reasonable fits to the data. The GLD estimate (dash-dotted curve) and the EWD estimate (dotted curve) are similar and fit the first class relatively better than those by EXP and WD. The right panel shows the Zipf plots of the four fitted distributions, where all four estimates fit the upper tail of the data well.

Table 3 summarizes the four fitted distributions. For Table 3, we find that

- EWD has the best fit to the upper tail of the data and has the smallest $\tilde{D}_n$ value. The $\tilde{D}_n$ value of GLD is larger than but close to that of EWD, and is much smaller than those by EXP and WD.

- EWD also provides the best fit to the whole data and has the smallest $\hat{D}_n$ value. GLD and WD also have smaller $\hat{D}_n$ values and provide overall good fits to the data.

**(a) Fitted Distribtions**

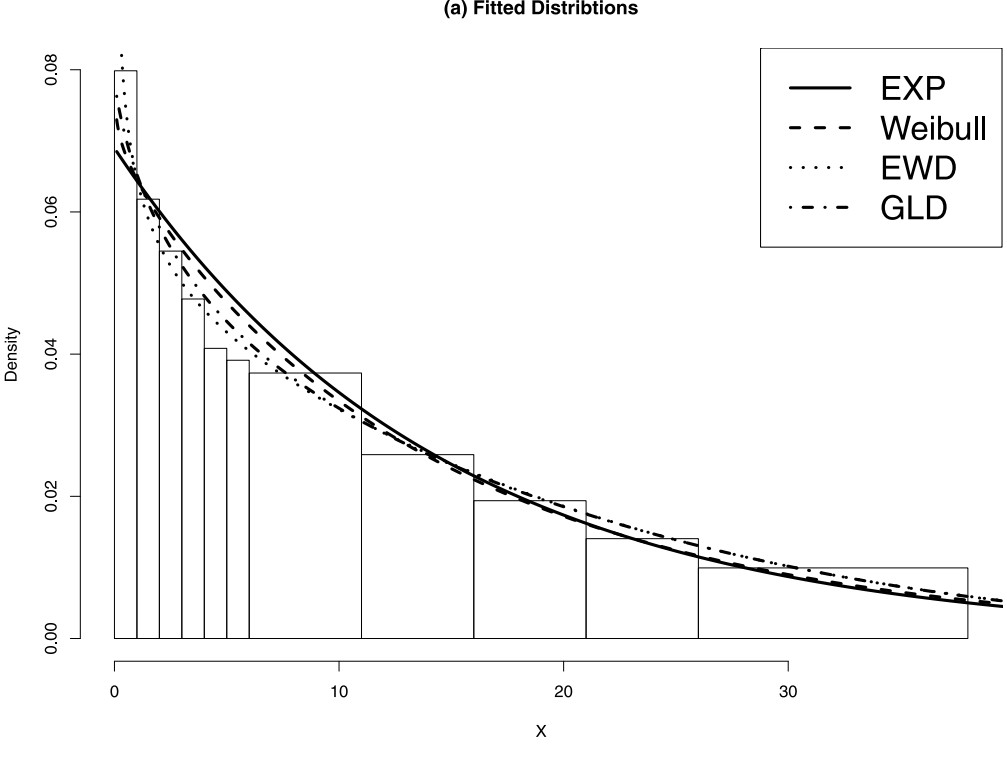

**(b) Zipf Plots of Fitted Distribtions**

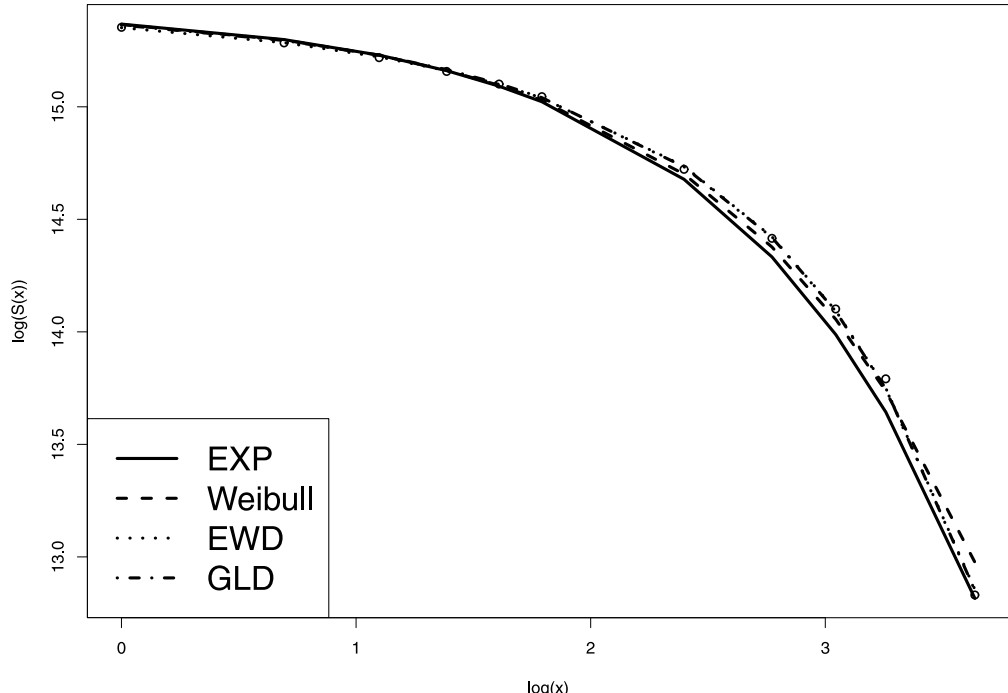

**Fig 2. Fitted distribution to 2014 US private-sector firm age data.** Left panel: estimated density functions. Right panel: Zipf plots of the fitted distributions.

**Table 3. Comparisons of fitted distributions to the 2014 firm age data.**

|  | $\tilde{D}_n$ | $\hat{D}_n$ | AIC | BIC | AICc |
|---|---|---|---|---|---|
| EXP | 0.1483 | 0.0285 | 24061198 | 24061211 | 24061198 |
| WD | 0.1455 | 0.0135 | 24046660 | 24046687 | 24046660 |
| EWD | 0.0418 | 0.0079 | 24005972 | 24006012 | 24005972 |
| GLD | 0.0445 | 0.0094 | 24012331 | 24012385 | 24012331 |

- EWD has the smallest AIC (Akaike information criterion), BIC (Bayesian information criterion), and AICc (AIC with a correction for small sample sizes) values.

Table 3 shows that each method has the same AIC and BIC values. This is because the sample size is very large ($n = 5, 058, 036$), and the the extra penalty term in AICc goes to zero. That says, AIC is equivalent to AICc for firm size and firm age data with very large sample sizes. The BIC is similar to AIC, but with a different penalty for the number of parameters ($p$). AIC has the penalty $2p$, whereas the penalty with BIC is $\log(n) \cdot p$. We propose to use BIC for model selection so preference will be given to distribution families with less parameters so as to avoid over-fitting to the data, which is especially important when the number of classes of the grouped data is small.

We further fit EXP, WD, EWD and GLD to the firm age data for years from 1983 to 2014 and compare their performances. The results are summarized in Table 4. Column 1 of Table 4

**Table 4. Comparisons of goodness-of-fit to firm age data.**

| Years (classes) | Dist | $\tilde{D}_n$ | $\hat{D}_n$ | Best($\tilde{D}_n$) | Best($\hat{D}_n$) | Best(BIC) |
|---|---|---|---|---|---|---|
| 1983-1987 (8) | EXP | 0.2988 ± 0.1003 | 0.1021 ± 0.0440 | 0 | 0 | 0 |
|  | WD | 0.0549 ± 0.0480 | 0.0354 ± 0.0315 | 0 | 0 | 0 |
|  | EWD | 0.0464 ± 0.0468 | 0.0308 ± 0.0305 | 2 | 2 | 2 |
|  | GLD | 0.0356 ± 0.0380 | 0.0219 ± 0.0222 | 3 | 3 | 3 |
| 1988-1992 (9) | EXP | 0.2988 ± 0.1657 | 0.0672 ± 0.0388 | 0 | 0 | 0 |
|  | WD | 0.0659 ± 0.0548 | 0.0326 ± 0.0264 | 0 | 0 | 0 |
|  | EWD | 0.0542 ± 0.0523 | 0.0257 ± 0.0252 | 1 | 0 | 1 |
|  | GLD | 0.0374 ± 0.0394 | 0.0165 ± 0.0168 | 4 | 5 | 4 |
| 1993-1997 (10) | EXP | 0.3156 ± 0.0979 | 0.0508 ± 0.0186 | 0 | 0 | 0 |
|  | WD | 0.0744 ± 0.0493 | 0.0260 ± 0.0174 | 0 | 0 | 0 |
|  | EWD | 0.0649 ± 0.0548 | 0.0212 ± 0.0164 | 1 | 2 | 2 |
|  | GLD | 0.0574 ± 0.0441 | 0.0188 ± 0.0115 | 4 | 3 | 3 |
| 1998-2002(11) | EXP | 0.3364 ± 0.1327 | 0.0421 ± 0.0129 | 0 | 0 | 0 |
|  | WD | 0.0822 ± 0.0461 | 0.0201 ± 0.0106 | 0 | 0 | 0 |
|  | EWD | 0.0710 ± 0.0569 | 0.0183 ± 0.0102 | 2 | 3 | 2 |
|  | GLD | 0.0681 ± 0.0480 | 0.0178 ± 0.0073 | 3 | 2 | 3 |
| 2003-2014(12) | EXP | 0.1928 ± 0.0656 | 0.0262 ± 0.0044 | 0 | 0 | 0 |
|  | WD | 0.1132 ± 0.0182 | 0.0165 ± 0.0033 | 0 | 0 | 0 |
|  | EWD | 0.0426 ± 0.0093 | 0.0105 ± 0.0006 | 9 | 6 | 10 |
|  | GLD | 0.0502 ± 0.0100 | 0.0121 ± 0.0033 | 3 | 6 | 2 |
| all | EXP | 0.2676 ± 0.1162 | 0.0508 ± 0.0351 | 0 | 0 | 0 |
|  | WD | 0.0858 ± 0.0438 | 0.0240 ± 0.0183 | 0 | 0 | 0 |
|  | EWD | 0.0529 ± 0.0399 | 0.0189 ± 0.0176 | 15 | 13 | 17 |
|  | GLD | 0.0499 ± 0.0330 | 0.0163 ± 0.0119 | 17 | 19 | 15 |

shows the years and the number of classes of the grouped firm age data. The means and standard deviations of $\tilde{D}_n$ and $\hat{D}_n$ are shown in columns 3 and 4, respectively. The last three columns show the number of years that each distribution outperforms the others in terms of $\tilde{D}_n$, $\hat{D}_n$ and BIC, respectively. Results show that

- EWD and GLD have overall better performances than EXP and WD. EWD has the best fits for 17 out of the 38 years firm age data, while GLD wins 15 times.

- GLD has smaller means and standard deviations of $\tilde{D}_n$ and $\hat{D}_n$ than EWD, respectively. WD also has reasonably good performances.

- The performances of all methods improve as the number of classes of the grouped data, $k$, increases. Both the means and standard deviations of $\tilde{D}_n$ and $\hat{D}_n$ decrease as $k$ increases. GLD has better performances than EWD when $k < 12$, and the performances of EWD are better than that of GLD when $k = 12$.

In summary, both GLD and EWD are good choices to model the firm age data. Although WD also has good performance in modeling the firm age data, it doesn't win in any year from 1983 to 2014. We also compare the performances of the above four methods in fitting the firm age data from 1979 to 1982, where the data have 4 to 7 classes, respectively. EWD wins for all four years. GLD is the second place winner for 1981 and 1982, and is not applicable for 1979 and 1980 due to the data having too few classes.

## Fitting firm size distributions

For the firm size data in Table 1, we take $b_0 = 0$ and assume that a firm in class "1-4" can have less than one employee. Such an approach is consistent with the grouping method for firm size in EU, where the firm size data are usually aggregated into five classes as "0-9", "10-19", "20-49", "50-249" and "250$^+$".

Plot (a) of Fig 3 shows the histogram of the 2014 US private-sector firm size data. Because the majority of firms have small sizes and the firm size spans over a very wide range, the histogram is severely skewed to the right. As shown in plot (b), the histogram is still right-skewed even when the firm sizes are log-transformed. Plot (c) shows the Zipf plot of the 2014 US private sector firm size data. It shows an obvious straight-line pattern. The Zipf plots of the US firm size data for each of the 38 years from 1977 to 2014 are shown in plot (d). All Zipf plots show parallel straight-line patterns. The slopes of the Zipf plots of the fitted Pareto distributions have a mean of -0.991 and a standard deviation of 0.013. The 95% confidence interval is $(-1.021, -1.004)$.

The type I Pareto distribution (PD) is a two-parameter power-law distribution. If $X$ is a random variable following a Pareto (Type I) distribution with shape parameter $\alpha > 0$ and scale parameter $x_m > 0$, $X$ has cumulative distribution function

$$F_{PD}(x, x_m, \alpha) = \begin{cases} 1 - (x_m/x)^{\alpha} & x \geq x_m, \\ 0 & x < x_m. \end{cases} \tag{14}$$

The maximum likelihood estimates (MLEs) of $\theta = (x_m, \alpha)$ can be computed numerically by maximizing the log-likelihood function in (1). It is easy to show that the Zipf plot of a Pareto random variable is a straight-line with slope $-\alpha$. Initial estimate of $\theta$ can be obtained by fitting a simple linear regression model to the Zipf plot of the grouped data (see plot (c) in Fig 3).

Because the location parameter $x_m$ has to be a positive value and we take $b_0 = 0$, we have to search for $x_m$ between the lower and upper limits of the first class with $b_0 < x_m \leq b_1$. In case

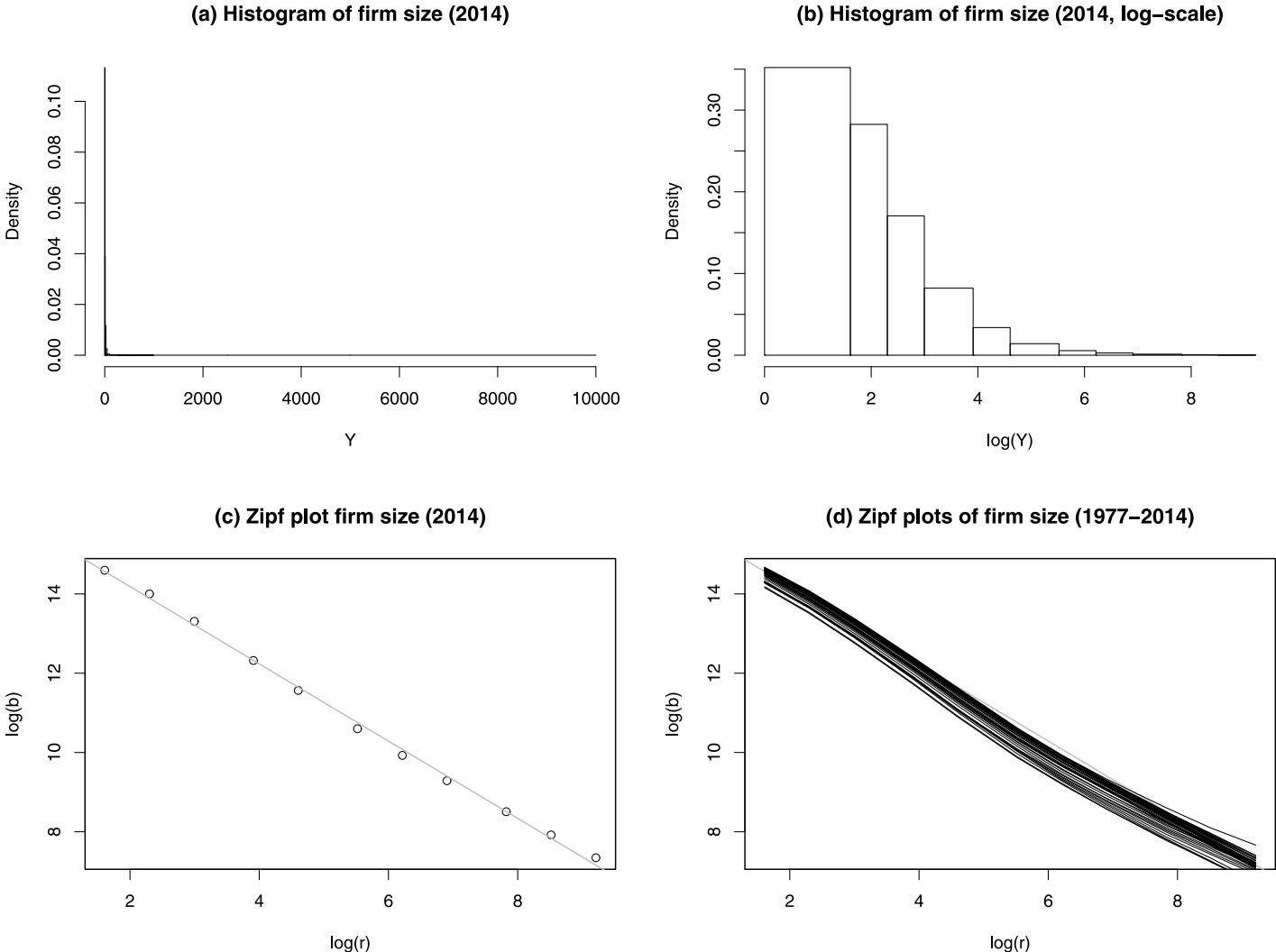

**Fig 3. US private-sector firm size data.** (a) Histogram of 2014 US firm size data. (b) Histogram of 2014 US firm size data after a logarithm transformation. To avoid infinite boundaries, we set $B'_0 = 1$. (c) Zipf plot of the 2014 US firm size data. The straight-line is the fitted least squares line to the Zipf plot. (d) Zipf plots of firm size data for years from 1977 to 2014.

$\hat{x}_m = b_1$, MLE of $\theta$ cannot be found based on the whole dataset, as the data in the first class need to be excluded. If that is the case, we use the least square estimate based on the Zipf plot to estimate $\alpha$ and find an MLE for $x_m$ by maximizing the likelihood in (1). The corresponding estimate of $x_m$ still needs be checked to make sure $0 < x_m < b_1$. We need to keep in mind that support of the PD is different from those by the other methods, and hence the performances of PD is not directly comparable with the others.

To ensure that the totality of data will be taken into account, we fit the generalized Pareto distributions (GPDs) to the grouped firm size data. The GPD is a family of continuous distributions with three parameters: location $\mu$, scale $\sigma > 0$ and shape $\xi$. The distribution function of $Y \sim GPD(\mu, \sigma, \xi)$ is

$$F_{GPD}(x, \mu, \sigma, \xi) = \begin{cases} 1 - [1 + \xi(x - \mu)/\sigma]^{-1/\xi}, & \text{for } \xi \neq 0, \\ 1 - e^{-(x-\mu)/\sigma}, & \text{for } \xi = 0. \end{cases} \tag{15}$$

For the firm size data, we fix $\mu = 0$ so no observation will be left out of the model fitting. When $\xi = 0$, GPD is reduced to an exponential distribution. With $\xi > 0$ and $\mu = \sigma/\xi$, the GPD is equivalent to PD with $x_m = \sigma/\xi$ and $\alpha = 1/\xi$. MLEs of $\sigma$ and $\xi$ can be found numerically by maximizing the log-likelihood function in (1).

As a conventional estimate, we fit the grouped firm size data to the log-normal distribution (LN) as well. If the data come from a LN with parameters, $\mu$ and $\sigma$, we have

$$\log(b_i) = \sigma \cdot z_i + \mu, \quad \text{for } i = 1, 2, \dots, k, \tag{16}$$

where $z_i$ is the quantile of a standard normal distribution with level $F(b_i)$ which can be approximated by $F_n(b_i)$. Therefore, we can fit a simple linear regression model of $\log(b_i)$ on $z_i$ and find initial estimate of $\mu$ and $\sigma$ using the y-intercept and slope of the first least squares line, respectively. MLE of $\theta = (\mu, \sigma)$ can be found numerically using (1).

Fig 4 shows the fitted distributions for the LN, PD, GPD and GLD for the 2014 US firm size data. The left panel shows that the estimated density curves by LN (solid), GPD (dotted) and GLD (dash-dotted) are close, while the estimate by PD is very different from the other three, which is mainly because PD assumes $X > \hat{x}_m$ and sets the density to zero for $X \leq \hat{x}_m$. The fitted density curves by PD, GPD and GLD stay very close for firm sizes larger than 10. The Zipf plots in the right panel show that PD (dashed curve) has the best fit to the upper tail of the data. Other than PD, GLD provides a very good fit to the upper tail of the data, and shows a straight-line pattern. Though, GLD slightly under-estimates the Zipf plot of data (and thus over-estimates the distribution function) for the top three classes. GPD also provides reasonable fit to the whole data and the upper tail while LN fails to model the upper tail of the firm size data.

The Zipf plots in plot (d) of Fig 3 demonstrate that the 1977-2014 firm size data have upper tail shapes of power law distributions. We fit the LN, PD, GPD and GLD to the 1977-2014 firm size data and compare their performances. Results in Table 5 show that GLD outperforms the other three estimators for all firm size data from 1977 to 2014 in terms of $\hat{D}_n$ and BIC. GLD has much smaller mean and standard deviation of $\hat{D}_n$ than those by the other three methods. PD does excellent job in modeling the upper tails of firm size data. It provides the best fits for all 38 years in terms of $\tilde{D}_n$—it has much smaller mean and standard deviation of $\tilde{D}_n$ than LN, GLD and GPD. PD also has slightly smaller $\hat{D}_n$ than GPD. However, due to the fact that PD cannot accommodate all firms in the first class when $b_0 = 0$, it should be used only if interests are not on the lower tail of the firm size distribution. Among all four methods, LN performs the worst.

## Fitting joint distribution firm size and firm age on 2014 US private sector firm data

To demonstrate the estimation of the joint distribution of firm age and firm size, we first fit marginal distribution of the firm age using EWD and GLD, respectively. Then we estimate the marginal distribution of firm size using GLD and GPD, respectively. Based on the two marginal distributions, we estimate the copula parameter and compute the joint density and distribution function of the bivariate distribution of firm age and firm size.

Fig 5 shows the contour plots of the estimated bivariate distributions using the four different combinations of the marginal distribution estimates for 2014 firm data as shown in Table 1. All four contour plots show that the firm age and firm size are positively correlated. The MLEs of $\Psi$ are 35.44, 34.83, 34.77 and 34.26 for plots (a)-(d), respectively. The four estimated joint distributions look very similar to each other.

**(a) Fitted Distributions**

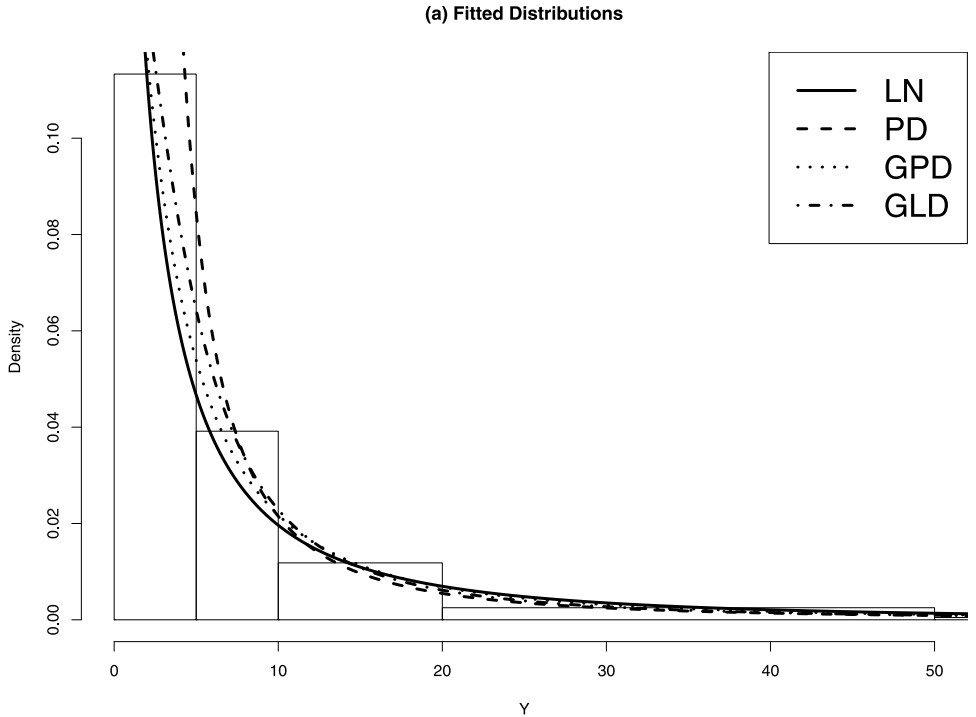

**(b) Zipf Plots of Fitted Distribtions**

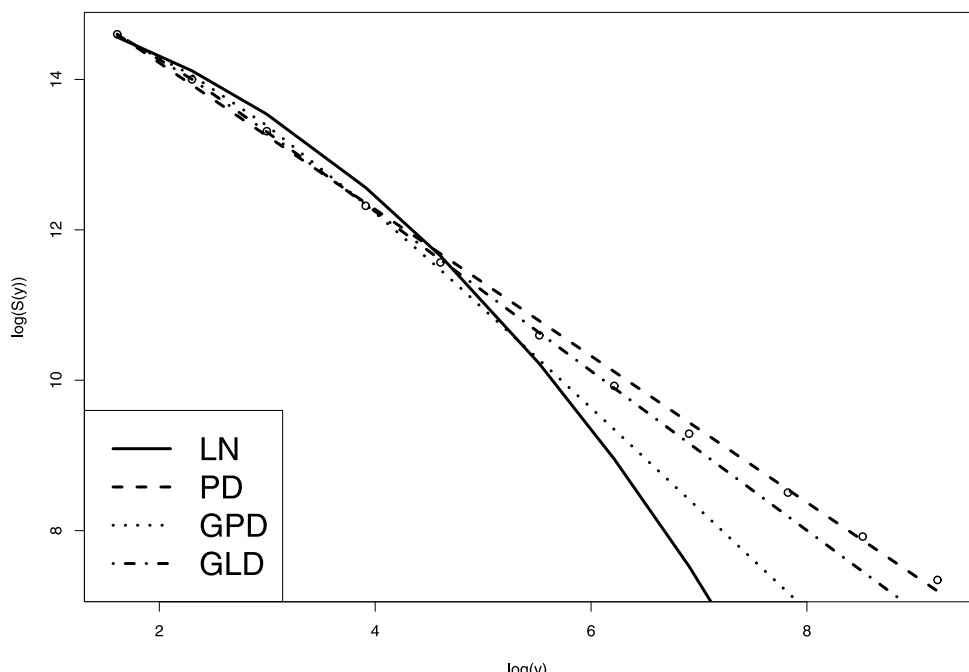

**Fig 4. 2014 US private-sector firm size data.** Left panel: estimated density functions. Right panel: Zipf plots of the fitted distributions.

**Table 5. Comparisons of goodness-of-fit to firm size data.**

| Dist | $\tilde{D}_n$ | $\hat{D}_n$ | Best($\tilde{D}_n$) | Best($\hat{D}_n$) | Best(BIC) |
|------|-----------|-----------|------------------|------------------|-----------|
| LN | 5.6550 ± 0.1193 | 0.0373 ± 0.0045 | 0 | 0 | 0 |
| PD | 0.2805 ± 0.0692 | 0.0151 ± 0.0042 | 38 | 0 | 0 |
| GPD | 2.5149 ± 0.2935 | 0.0165 ± 0.0020 | 0 | 0 | 0 |
| GLD | 0.7443 ± 0.1339 | 0.0013 ± 0.0003 | 0 | 38 | 38 |

In Fig 6, we show the contour plot of the fitted joint distribution by excluding the data in the first class of firm size. We keep all age groups and estimate the marginal distribution of firm age with EWD. Without the first class, GLD fits the firm size data better than PD and GPD in terms of BIC and $\hat{D}_n$, but PD fits the upper tail the best (with the smallest $\tilde{D}_n$). In Fig 6, the two estimates of the joint distribution are similar. The MLEs of $\Psi$ are 43.19 and 53.12 for

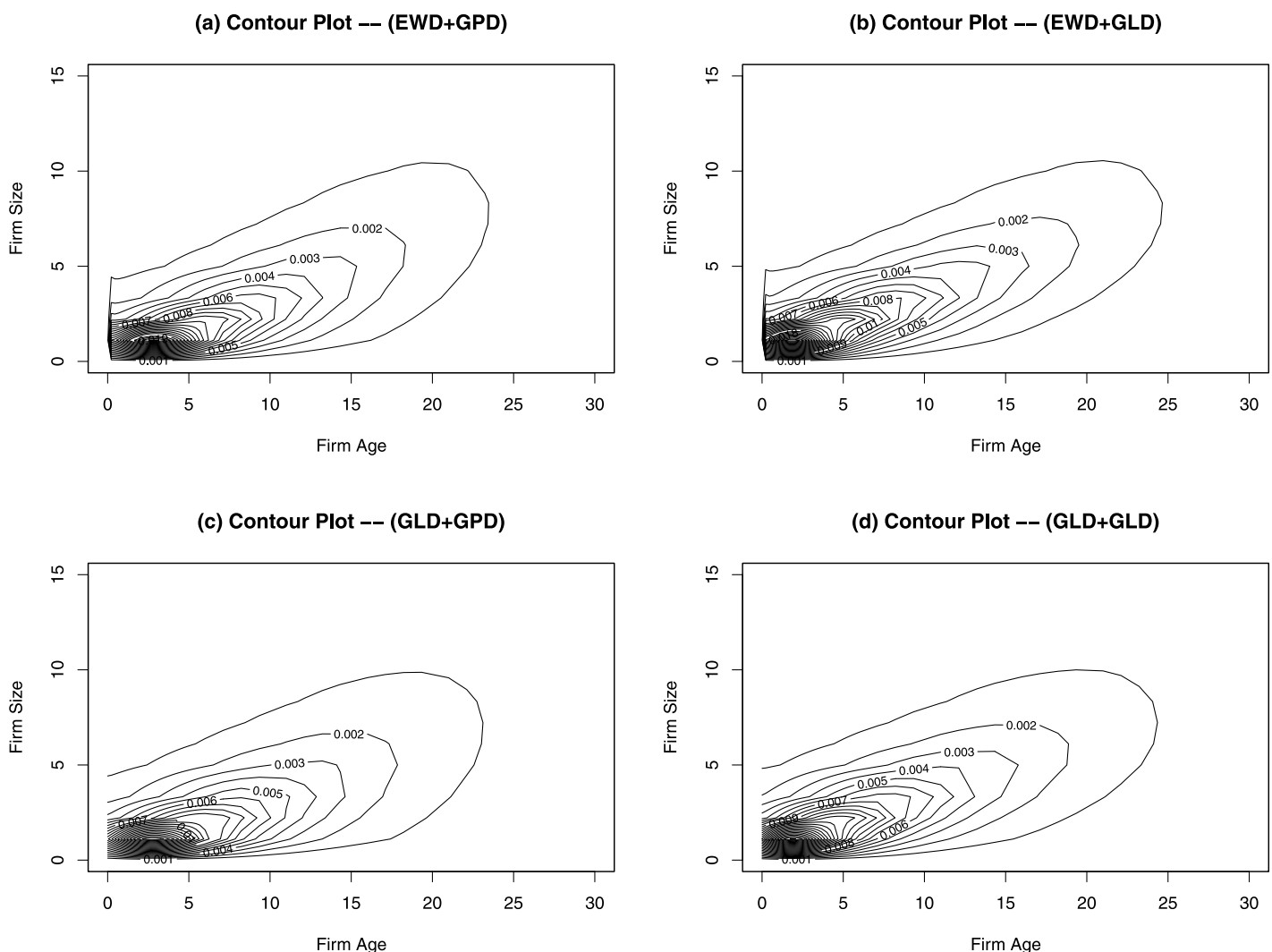

**Fig 5. Estimated joint distribution of firm age and firm size for 2014 US private-sector firms.** Plots (a) and (b), the marginal distribution of firm age is estimated by using EWD. Plots (c) and (d), the marginal distribution of firm age is estimated by using GLD. Plots (a) and (c), the marginal distribution of firm size is estimated by using GPD. Plots (c) and (d), the marginal distribution of firm size is estimated by using GLD.

**(d) Contour Plot –– (GLD+GLD)**

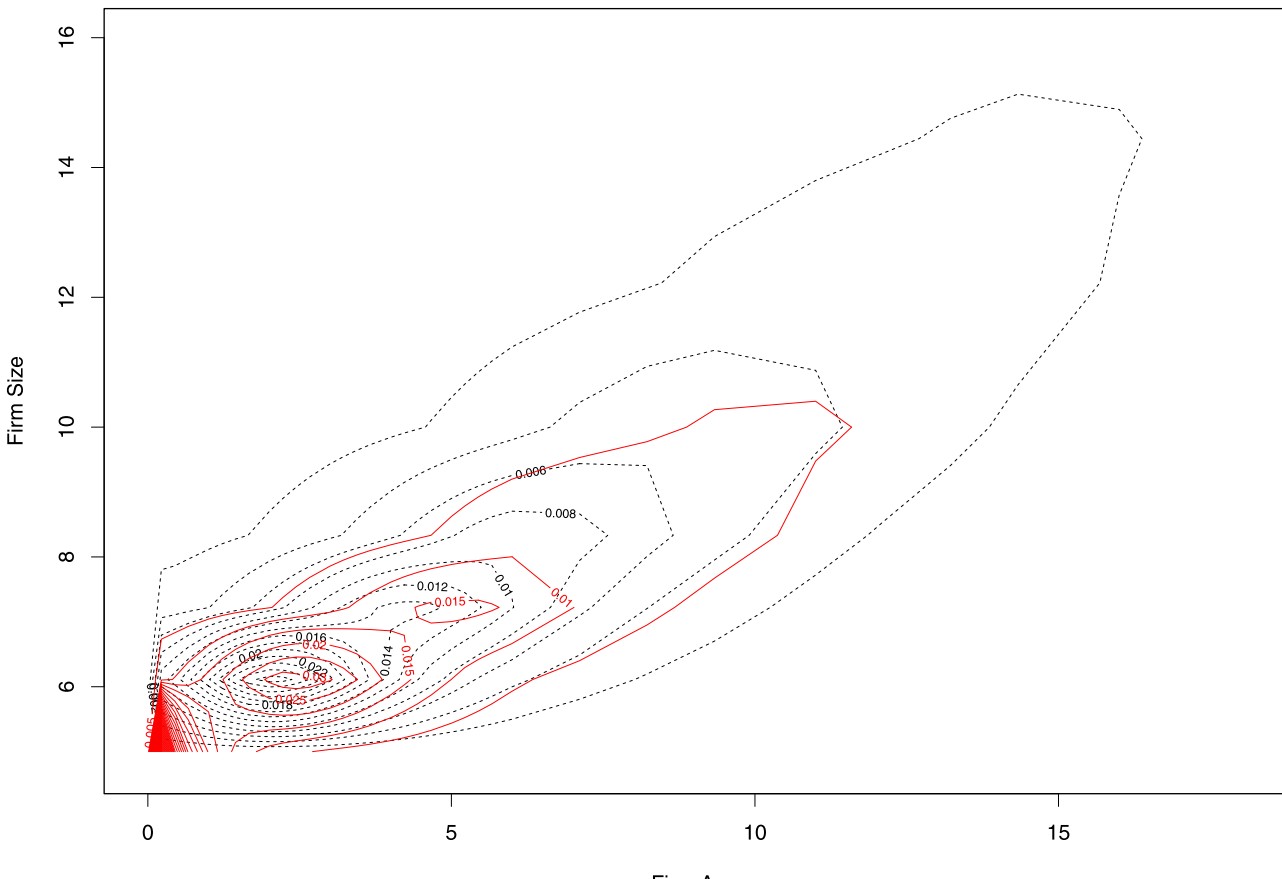

**Fig 6. Estimated joint distribution of firm age and firm size for 2014 US private-sector firms without the first class.** The dashed curves are the estimates using EWD for firm age and GLD for firm size. The solid red curves are the estimates using EWD for firm age and PD for firm size.

GLD and PD, respectively. That says, without taking account of the firms with size less than 5, the association between firm size and firm age becomes stronger. This makes sense because the firm class contains 56.66% of the firms and the MLE of $\Psi$ is likely to under-estimate the association by assuming all these firms having the same ranks in terms of age and size.

## Conclusion

Firm size and/or firm age data are usually released as pre-binned top-coded data. Due to the fact that the data are heavily aggregated, non-parametric methods such as kernel smoothing won't work well. When the data are top-coded, the distributions fitted using non-parametric methods usually lack the capability of predicting beyond the scope of the observed data. GLD is parametric and very flexible in shapes and tails, and hence depends less on the assumption of the functional form of the underlying distributions. Results have shown that GLD has very good performances in modeling the firm age and firm size data. The firm data are usually very large so the quantile levels of the class boundaries are well approximated. This ultimately improves the reliability of the GLD estimates. In addition, due to the firm data being grouped, the computational burden in fitting the GLDs will be greatly reduced. For the US private section firm size data, PD might be a good choice due to its simpler form than GLD. However,

when the firm size is measured using some other variables instead of the number of employees, the firm size may not follow a power law and PD might not be appropriate. For some extreme cases, GPD might not be appropriate as well. On the contrary, GLD works for many data with different shapes and tails. Other than GLD, EWD can also be selected to model firm size data.

The estimation of the copula parameter doesn't depend on the methods used to estimate the marginal distribution. If the research interests focus on the lower tails of the distributions, some non-parametric methods, such as the kernel density estimators, can also be used. The non-parametric methods are free of distributional assumptions and can model the data locally very well. If the upper boundary of the last class is finite and well defined, the non-parametric estimators are also good choices. However, if the upper tail behaviors are of interest, the parametric methods are preferred, as we can predict beyond the scopes of the data with the fitted model.

For data that are severely skewed to the right, such as firm size data, the goodness-of-fit needs to be checked carefully. The best fit can be chosen differently depending on the objectives of a study. If the upper tail behaviors are of interest, the goodness-of-fit shall be checked visually using the Zipf plot and/or using $\tilde{D}_n$. The goodness-of-fit of the distribution over the whole range of data can be checked using the KS statistic $\hat{D}_n$, or a Chi-square test as needed. BIC is believed to be a better choice than AIC and AICc as it gives more penalty to distributions with more parameters and hence avoid over-fitting to the grouped data, which eventually will help to improve the predictability of the fitted model.

As a limit, the GLD method may not work well when the number of classes is very small, say 6 or less. However, we need to keep in mind that not too many options are available when little information is available. In such cases, a family of distributions with fewer parameters are recommended to estimate the marginal distributions and the corresponding joint distributions.

When longitudinal firm data are available, the estimated distributions can be incorporated into regression models to take into consideration information of other covariates to study the firm growth or firm survivability. Yang et al. used several machine learning models to evaluate the influence of periodic components on short-term speed prediction based on aggregated data [51]. In our future studies, efforts will be devoted to simultaneously modeling firm size and age data of a sequence of multiple years to investigate the change of the associations between firm size and firm age in terms of the Plackett copula parameter.

All the algorithms have been implemented in the CRAN R package `bda` v14.3.15 or higher. Sample codes and technical details can be found from the help manual for the command *fit. FSD*.

## Author Contributions

**Conceptualization:** Shu-Guang Zhang, Bin Wang.

**Formal analysis:** Bin Wang.

**Investigation:** Bin Wang.

**Methodology:** Shu-Guang Zhang, Bin Wang.

**Resources:** Chen Ge, Shu-Guang Zhang, Bin Wang.

**Software:** Bin Wang.

**Supervision:** Shu-Guang Zhang.

**Validation:** Chen Ge.

**Visualization:** Bin Wang.

**Writing – original draft:** Chen Ge, Bin Wang.

**Writing – review & editing:** Chen Ge, Shu-Guang Zhang, Bin Wang.

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
