## [Decision Letter · Decision Letter 0]

21 May 2020

PONE-D-20-08858

Modeling the Joint Distribution of Firm Size and Firm Age Based on Grouped Data

PLOS ONE

Dear Dr. Ge,

Thank you for submitting your manuscript to PLOS ONE. After careful consideration, we feel that it has merit but does not fully meet PLOS ONE’s publication criteria as it currently stands. Therefore, we invite you to submit a revised version of the manuscript that addresses the points raised during the review process.

We would appreciate receiving your revised manuscript by Jul 05 2020 11:59PM. To enhance the reproducibility of your results, we recommend that if applicable you deposit your laboratory protocols in protocols.io, where a protocol can be assigned its own identifier (DOI) such that it can be cited independently in the future. For instructions see: http://journals.plos.org/plosone/s/submission-guidelines#loc-laboratory-protocols

We look forward to receiving your revised manuscript.

Kind regards,

Yajie Zou

Academic Editor

PLOS ONE

Journal Requirements:

Reviewers' comments:

Reviewer's Responses to Questions

**Comments to the Author**

1. Is the manuscript technically sound, and do the data support the conclusions?

Reviewer #1: Yes

Reviewer #2: Yes

2. Has the statistical analysis been performed appropriately and rigorously? 

Reviewer #1: Yes

Reviewer #2: Yes

3. Have the authors made all data underlying the findings in their manuscript fully available?

Reviewer #1: Yes

Reviewer #2: Yes

4. Is the manuscript presented in an intelligible fashion and written in standard English?

Reviewer #1: Yes

Reviewer #2: Yes

5. Review Comments to the Author

Reviewer #1: In general, the methods developed in this paper were sound and clear. They investigated multiple parametric methods for firm size and firm age modeling based on grouped data, and proposed a method to estimate the joint distribution of firm size and firm age. The applications of the methods seem interesting.

However, I felt the paper structure was somehow dis-organized.

I suggested the authors have five sections clearly:

(1) Introduction

(2) Materials/Datasets: describe the Legacy BDS Firm Size and Age Data

(3) Methods:

(3.1) Marginal Distribution Estimation Based on Grouped Data

(3.2) Estimating The Joint Distribution of Firm Size and Firm Age

(4) Results

(5) Conclusion

The detailed analysis results in "Estimating The Marginal Distribution of Firm Age" and "Estimating The Marginal Distribution of Firm Size" should be put in the "Results" section.

Reviewer #2: This paper jointly modeled the firm size and firm age. Some conclusions and phenomena were found. This paper’s topic is interesting, but there are some aspects need to be improved from its current form:

1. The relevant studies in the literature review are not enough for current studies about the topic, some machine learning approaches may be also appropriate for this dataset. For example, see: Evaluation of Short-Term Freeway Speed Prediction Based on Periodic Analysis Using Statistical Models and Machine Learning Models. Journal of Advanced Transportation, 2020.

2. The problem description section should be added, and the main contribution should be further clearly presented.

3. There are some typos in the paper, for example, see Equation (15).

4. In the results and discussions section, I think that the authors should add more discussion to show the practical significance of the proposed research.

5. None of the explanatory variables are considered in this study to develop the bivariate copula regression model. This issue should also be discussed and what are the potential variables to be included for future research.

Therefore, I think this paper is interesting and the above aspects should be further improved.

6. PLOS authors have the option to publish the peer review history of their article (what does this mean?). If published, this will include your full peer review and any attached files.

Reviewer #1: No

Reviewer #2: No

---

## [Author Response · Author response to Decision Letter 0]

29 May 2020

5. Review Comments to the Author

Reviewer #1: In general, the methods developed in this paper were sound and clear. They investigated multiple parametric methods for firm size and firm age modeling based on grouped data, and proposed a method to estimate the joint distribution of firm size and firm age. The applications of the methods seem interesting.

However, I felt the paper structure was somehow dis-organized.

I suggested the authors have five sections clearly:

(1) Introduction

(2) Materials/Datasets: describe the Legacy BDS Firm Size and Age Data

(3) Methods:

(3.1) Marginal Distribution Estimation Based on Grouped Data

(3.2) Estimating The Joint Distribution of Firm Size and Firm Age

(4) Results

(5) Conclusion

The detailed analysis results in "Estimating The Marginal Distribution of Firm Age" and "Estimating The Marginal Distribution of Firm Size" should be put in the "Results" section.

RESPONSE: 1) We thank the reviewer for the valuable suggestion on restructuring the manuscript. In the revised version, we created a new section named “Materials and dataset” to describe the legacy DBS firm size and age data. The statistical methodology has been kept in a new section named “Methods”. 2) To make the structure more organized, we removed the section title “Marginal Distribution Estimation Based on Grouped Data” and reduced the depth of the subsections of “Fitting A Parametric Model to Grouped Data” and “Goodness-of-fit Assessment” in “Methods” section. The two sub-sections "Estimating The Marginal Distribution of Firm Age" and "Estimating The Marginal Distribution of Firm Size" have been merged with the corresponding subsections in “Results” section.

Reviewer #2: This paper jointly modeled the firm size and firm age. Some conclusions and phenomena were found. This paper’s topic is interesting, but there are some aspects need to be improved from its current form:

1. The relevant studies in the literature review are not enough for current studies about the topic, some machine learning approaches may be also appropriate for this dataset. For example, see: Evaluation of Short-Term Freeway Speed Prediction Based on Periodic Analysis Using Statistical Models and Machine Learning Models. Journal of Advanced Transportation, 2020.

RESPONSE: Thanks for the valuable comments. We reviewed the paper by Yang et al (2020) and some cited articles. Discussion on the use of machine learning methods to evaluate the impact of periodic component of time has been added to Discussion section.

2. The problem description section should be added, and the main contribution should be further clearly presented.

RESPONSE: Per the reviewer’s recommendation, description of the problem/ statistical challenges have been discussed and added to the “Introduction” section. More comments on the main contribution have been discussed in both the “Abstract” and the “Conclusion” section. Thanks.

3. There are some typos in the paper, for example, see Equation (15).

RESPONSE: The typo in Equation (15) has been corrected. In addition, the revised manuscript has been carefully reviewed and proofread by a native English-speaker.

4. In the results and discussions section, I think that the authors should add more discussion to show the practical significance of the proposed research.

RESPONSE: The main contributions including the practical significance have been discussed in the “Conclusion” section.

5. None of the explanatory variables are considered in this study to develop the bivariate copula regression model. This issue should also be discussed and what are the potential variables to be included for future research.

RESPONSE: In this manuscript, we focus on univariate and bivariate distribution estimations based on grouped top-coded data. It is a great idea to take into consideration of other explanatory variables in statistical inferences. We briefly discuss this as our potential future research topic. Thanks.

---

## [Decision Letter · Decision Letter 1]

12 Jun 2020

Modeling the Joint Distribution of Firm Size and Firm Age Based on Grouped Data

PONE-D-20-08858R1

Dear Dr. Ge,

We’re pleased to inform you that your manuscript has been judged scientifically suitable for publication and will be formally accepted for publication once it meets all outstanding technical requirements.

Kind regards,

Yajie Zou

Academic Editor

PLOS ONE

Reviewers' comments:

Reviewer's Responses to Questions

**Comments to the Author**

1. If the authors have adequately addressed your comments raised in a previous round of review and you feel that this manuscript is now acceptable for publication, you may indicate that here to bypass the “Comments to the Author” section, enter your conflict of interest statement in the “Confidential to Editor” section, and submit your "Accept" recommendation.

Reviewer #1: All comments have been addressed

Reviewer #2: All comments have been addressed

2. Is the manuscript technically sound, and do the data support the conclusions?

Reviewer #1: Yes

Reviewer #2: Yes

3. Has the statistical analysis been performed appropriately and rigorously? 

Reviewer #1: Yes

Reviewer #2: Yes

4. Have the authors made all data underlying the findings in their manuscript fully available?

Reviewer #1: Yes

Reviewer #2: Yes

5. Is the manuscript presented in an intelligible fashion and written in standard English?

Reviewer #1: Yes

Reviewer #2: Yes

6. Review Comments to the Author

Reviewer #1: The authors have addressed my questions. The paper has been organized and is more readable now. I have no further comments.

Reviewer #2: I think this paper is interesting. The authors have adequately addressed my comments and some aspects mentioned have been further improved.

7. PLOS authors have the option to publish the peer review history of their article (what does this mean?). If published, this will include your full peer review and any attached files.

Reviewer #1: No

Reviewer #2: No

---

## [Editor Report · Acceptance letter]

17 Jun 2020

PONE-D-20-08858R1 

Modeling the Joint Distribution of Firm Size and Firm Age Based on Grouped Data 

Dear Dr. Ge:

I'm pleased to inform you that your manuscript has been deemed suitable for publication in PLOS ONE. Congratulations! Your manuscript is now with our production department. 

Kind regards, 

on behalf of

Dr. Yajie Zou 

Academic Editor

PLOS ONE